



# Evaluating spatiotemporal variations and exposure risk of ground-level ozone concentrations across China from 2000 to 2020 using satellite-derived high-resolution data

Qingqing He[1, 2, *], Jingru Cao[1], Pablo E. Saide[2, 3], Tong Ye[1], Weihang Wang[1]

[1] School of Resource and Environmental Engineering, Wuhan University of Technology, Wuhan 430070, China

[2] Department of Atmospheric & Oceanic Sciences, University of California, Los Angeles, Los Angeles, California 90095, United States

[3] Institute of the Environment and Sustainability, University of California, Los Angeles, Los Angeles, California 90095, United States

*Correspondence to*: Qingqing HE (qqhe@whut.edu.cn)

**Abstract:** Understanding the spatial and temporal characteristics of both long- and short-term exposure to ground-level ozone is crucial for refining environmental management and improving health studies. However, such studies have been constrained by the availability of spatiotemporal high-resolution data. To address this gap, we characterized ground-level ozone variations and exposure risks across multiple spatial (pixel, county, region, and national) and temporal (daily, monthly, seasonal, and annual) scales using daily 1-km ozone data from 2000 to 2020, derived from satellite land surface temperature data via a machine-learning hindcast method. The model provided reliable estimates, validated through rigorous cross-validation and direct comparison with external ground-level ozone measurements. Our long-term estimates revealed seasonal shifts in high-exposure ozone centers: spring in eastern China, summer in the North China Plain (NCP), and autumn in the Pearl River Delta (PRD). A non-monotonous trend was observed, with ozone levels rising from 2001-2007 at a rate of 0.47 µg/m³/year, declining after 2008 (-0.58 µg/m³/year), and increasing significantly from 2016-2020 (1.16 µg/m³/year), accompanied by regional and seasonal fluctuations. Notably, ozone levels increased by 0.63 µg/m³/year in summer in the NCP during the second phase, and by 6.38 µg/m³/year in autumn in the PRD during the third phase. Exposure levels over 100 µg/m³ have shifted from June to May, and levels exceeding 160 µg/m³ were primarily seen in the NCP, showing an expanding trend. Our day-to-day analysis highlights the influence of meteorological factors on extreme events. These findings emphasize the need for increased public health awareness and stronger mitigation efforts.

## 1 Introduction

Ground-level ozone is a critical pollutant and greenhouse gas in the atmosphere. A growing body of research has demonstrated that both short-term and long-term exposure to ambient ozone are linked to various adverse health outcomes, including asthma (Nicholas et al. 2020), respiratory tract infections (Burnett et al. 1994), and even premature deaths (Maji and Namdeo 2021). Moreover, severe ozone pollution in the ambient environment also impacts agriculture crops and contributes to climate change (Li et al. 2018; Ramya et al. 2023). Therefore, it is crucial to investigate the long-term variation of ground-level ozone, especially for China, a country undergoing significant atmospheric environmental changes due to its rapid economic growth and evolving air pollution control policies over the last two decades. Additionally, influenced by a mix of meteorological conditions, local emissions, and regional transport mechanisms (Fiore et al. 2003; Jaffe 2011; Monks et al. 2015), ground-level ozone exhibits considerable



heterogeneities in its spatial distribution and temporal trend. Understanding these fine-scale variations can provide more precise information about local ozone variations, for example, helping to identify local ozone spikes (Shi et al. 2023; Youliang Chen 2022) and allowing for accurate assessments of human exposure to ozone at the community or even neighborhood level (Alexeeff et al. 2018). However, such intricate tasks cannot be accomplished solely by the ground-level air quality monitoring network. While

monitoring networks offer accurate ozone concentration data, their limited observation duration and sparse station distribution inadequately capture intraurban variations, often resulting in underestimates of neighborhood and individual exposure variability (Dias and Tchepel 2018). Therefore, it is necessary to enhance the understanding in ground-level ozone variation and enable more effective mitigation measures using full-coverage, long-term ozone data with high spatiotemporal resolution.

To date, various methods have been employed to address the limitations of ground-level ozone data for a more comprehensive understanding. Atmospheric chemical transport models (CTMs) have been extensively used to simulate ground-level ozone concentrations (Sharma et al. 2017). However, this method typically provides coarse-resolution simulations (usually ≥12km×12km) (Qiao et al. 2019; Sun et al. 2019), due to its high computation cost, and its accuracy needs to be improved due to the large uncertainty in the emission inventory and many assumptions made when running the CTM model (Sharma et al. 2017). Advanced

statistical/machine-learning algorithms provide an alternative way to obtain spatiotemporal patterns in ozone. By combining with ground-level ozone observations and satellite-retrieved columnar ozone and/or precursor data, those machine-learning methods have significantly improved estimation accuracies (e.g., validated $R^2$ values higher than 0.80) and refined the spatial resolution of the estimates (e.g., $0.1° \times 0.1°$ and $0.05° \times 0.05°$) (Chen et al. 2021; Li et al. 2020; Li and Cheng 2021; Mu et al. 2023a; Mu et al. 2023b; Xue et al. 2020; Zhang et al. 2020; Zhu et al. 2022). Given that the variation in ground-level ozone is influenced by

atmospheric and geographic factors (Fu and Tai 2015; Li et al. 2019; Tu et al. 2007; Wang et al. 2022b; Zhu et al. 2022), several studies have employed statistical and machine-learning algorithms using atmospheric components (e.g., $PM_{2.5}$), surface conditions (e.g., elevation), and meteorological factors (e.g., temperature, wind, sunshine, and precipitation), along with relatively high-resolution data, as predictors for modeling. While previous studies have produced ground-level ozone concentration estimates across China with improved spatial resolutions (Chen et al. 2021; Liu et al. 2020; Ma et al. 2022a; Wei et al. 2022; Zhan et al.

2018), there remain at least two key limitations: (1) Despite advancements, these studies have not incorporated suitable proxies with high spatiotemporal resolution data in their models, which are necessary to capture fine-scale ozone gradients. As a result, the spatial resolutions of these estimates remain relatively coarse (e.g., $0.0625° \times 0.0625°$ and $0.1° \times 0.1°$); (2) Although full-coverage ozone estimates have been provided, few studies track long-term variations before 2005, and even fewer offer external or independent validation for pre-2013 estimates when national air quality monitoring data were unavailable. Consequently, the

current datasets are insufficiently detailed or validated to detect fine-scale intra- and inter-city ozone variations over time, thereby limiting the accuracy of exposure assessments.

Ozone is a short-lived pollutant, exhibiting significant spatial and temporal variations even over small areas and short periods (Mukherjee et al. 2018; Shi et al. 2023). The scarcity of long-term, spatiotemporally detailed ozone data has historically confined

ozone research to identifying exposure hotspots and events from a broad-scale or a time-aggregated perspective (Liu et al. 2022; Mashat et al. 2020; Xia et al. 2022). The detailed intra-urban differences and short-duration phenomena over the past two decades remain largely unexplored. To address this gap, our study utilizes a long-term ground-level ozone concentration dataset across China from 2000 to 2020 with daily, 0.01° (~1 km) spatiotemporal resolution. This dataset is used to evaluate general spatial



patterns of long-term ozone variations, identify hotspots of population exposure to ground-level ozone across multiple spatial and temporal scales, and examine the implications for mitigation policies and public health. The ozone dataset is estimated using our previously developed spatiotemporal high-resolution machine learning-based ozone estimation framework, which incorporates land surface temperature (LST), derived from long-term, high-resolution satellite remote sensing observations, as a primary predictor (He et al. 2024). To ensure the reliability of the long-term exposure analysis, the estimates were evaluated through rigorous cross-validation and independently validated using external ozone measurements. The exposure analysis integrates these

high-resolution ozone estimates with detailed population distributions derived from geographic big data.

## 2 Data and Methodology

### 2.1 Long-term ozone estimates and validation

The present study builds on our previously developed high-resolution ozone modeling framework to hindcast long-term ozone

concentration data across China from 2000 to 2020. That framework was designed to predict the daily maximum 8-hour average (MDA8) ozone concentrations at a 0.01° spatiotemporal resolution using the extreme gradient boosting (XGBoost) algorithm. It incorporated four groups of predictors: meteorological parameters (e.g., land surface temperature (LST), boundary layer height), pollutant variables (e.g., nitrogen dioxide, aerosol optical depth), geographical covariates (e.g., elevation, land cover classification), and temporal dummy variables (e.g., day of the year). In that model, satellite-derived LST data, with full coverage and daily, 0.01°

resolution, served as the primary predictor. Since the data sources, preprocessing approaches, and predictor selection for the current long-term hindcast estimation model closely follow the previously developed modeling framework, further details are documented in Text S1.

The model development process closely mirrors that of our previous high-resolution model. The XGBoost algorithm (Chen and

Guestrin 2016) is also utilized to train the long-term hindcast model due to its demonstrated effectiveness in ground-level ozone estimation at an acceptable computational cost, as indicated by our previous study (Li et al. 2024b). Given that the Chinese National Air Quality Monitoring Network (NAQMN) was not established before 2013 and monitoring data from that period is unavailable, we apply a widely-used pre-2013 $PM_{2.5}$ hindcast modeling approach to predict long-term ozone concentrations (Ma et al. 2022b). Specifically, we train the ozone estimation model on data from 2014 to 2020, and once the model is adequately trained, we apply

it to retrospectively predict ozone concentrations for the past two decades, including the 14 years preceding the establishment of the NAQMN. The study period is partitioned following the approach of a previous study (Zhu et al. 2022), with 2014-2020 as the training period and 2000-2013 as the hindcast period. We exclude the year 2013 from our hindcast modeling due to the limited number and data quality of air quality monitoring stations during NAQMN's inaugural year. We focus on optimizing four critical hyperparameters of XGBoost to balance model performance and computational efficiency: (1) n_estimators, the number of trees

in the model; (2) max_depth, which controls the maximum tree depth to prevent overfitting; (3) colsample_bytree, the proportion of features sampled for each tree; and (4) min_child_weight, the minimum number of samples required in a child node. We employ a random search with cross-validation to find the optimal settings for these hyperparameters, which are set at 400, 14, 0.8, and 4, respectively. This hyperparameter setting is a tradeoff between model performance and the computational demand. We implemented the modeling process in Python (ver. 3.9) with the Sklearn XGBoost package (ver. 1.7.3).



Due to the absence of nationwide ground-level ozone measurements prior to 2013, directly assessing the estimates for those years is challenging. To address this, we employ the leave-one-year-out cross-validation (CV) method to evaluate the reliability of our long-term hindcast model in estimating years without ground-level ozone measurements. This approach involves withholding data from one entire year during model training, simulating a hindcast scenario where ozone measurements are unavailable. This state-

of-the-art evaluation technique is widely used in PM$_{2.5}$ hindcast modeling for pre-2013 predictions (Ma et al. 2022b). Additionally, although limited, some ground-level ozone measurements from before 2013 are available from monitoring sites in Hong Kong. To further validate the pre-2013 predictions, we use these independent Hong Kong ozone measurements—excluded from model development—to directly assess the model's performance during the extended historical period. This provides a more straightforward evaluation, given the lack of nationwide pre-2013 ozone data. In addition to validating hindcast predictions for the

pre-2013 period, we also apply the random 10-fold CV to assess the overall performance of our model. This process involves randomly dividing the sample dataset into 10 subsets, using nine subsets to train the model and the remaining subset to test it. This procedure is repeated 10 times to ensure that each daily MDA8 measurement has a corresponding estimate for comparison. The site- and day-based CVs specifically assess the model's spatial and temporal performance. We compute several statistical metrics, including R², RMSE (Root-Mean-Squared Error), and MAE (Mean Absolute Error), to compare the MDA8 measurements with

the model estimates.

### 2.2 Multi-scale spatiotemporal analysis

We generated full-coverage, daily 1-km resolution ozone estimates across China from 2000 to 2020 using the proposed hindcast machine-learning method. Based on these long-term, high-resolution spatiotemporal estimates, we analyzed interannual, seasonal,

and monthly variations, as well as short-term exposure characteristics, at national, regional, county, and pixel scales. Particular attention was given to typical high-exposure regions, which were identified by mapping the spatial distributions of seasonal averages across the study areas over the past two decades.

### 2.2.1 Long-term trend analysis

To assess long-term exposure trends, we combined the MDA8 ozone estimates with concurrent yearly 1-km LandScan population

distributions (Rose et al. 2020) to compute the annual and seasonal population-weighted mean MDA8 ozone concentrations for China and typical regional hotspots from 2000 to 2020. These population-weighted concentrations were used to analyze interannual variations, seasonal fluctuations, and regional differences in long-term ozone trends at both national and regional levels. The four seasons were defined as follows: spring (March-May), summer (June-August), autumn (September-November), and winter (December-February). The detailed formulation for calculating population-weighted ozone levels ($O_3\_POP$) for a given region is

presented in Eq. 1. The long-term linear trend was estimated using the least-squares approach, consistent with previous studies (He et al. 2016; Li et al. 2019).

$$O_3\_POP = \sum(POP_i \times O_{3_i})/\sum POP_i \qquad \text{Eq. 1}$$

where is $POP_i$ and $O_{3_i}$ denote the population and the estimated MDA8 O$_3$ level in grid cell $i$.



### 2.2.2 Monthly pattern analysis

We calculated the monthly population-weighted mean MDA8 ozone concentrations from 2000 to 2020 and identified the peak and trough values from these monthly time series. To capture both seasonal extremes and the underlying background ozone concentrations, we calculated linear trends separately for the peaks and troughs. Additionally, we applied a Mann-Kendall test to the monthly peak time series to determine whether there is a statistically significant trend in maximum ozone concentrations over the two decades for various regions in China. To assess the extent of severe ozone pollution across counties over time, we generated

time series data on the number of counties with monthly ozone concentrations exceeding 100 μg/m³ and analyzed the linear trend. The threshold of 100 μg/m³ was selected based on the Chinese National Air Quality Standard Level 2 and the WHO (World Health Organization) air quality guideline as an indicator of severe exposure. From the monthly exposure time series, we selected a month with severe ozone pollution to map the spatial disparity in ozone exposure at the county level.

### 2.2.3 Short-term characteristics analysis

By overlaying the daily ozone estimates with 1-km LandScan population maps, we calculated the number of people exposed to different ozone concentration levels. Our primary focus was on two key thresholds: 100 μg/m³, the 8-hour air quality guideline recommended by the WHO, and 160 μg/m³, the Level 2 standard set by the Chinese National Air Quality Standard. Additionally, we generated a high-exposure risk map for extremely severe ozone pollution by showing the spatial distribution of the percentage of days with ozone concentrations exceeding 160 μg/m³, the second level of the national air quality standard.

## 3 Results

### 3.1 Evaluation results of model performance and predictions

### 3.1.1 Validation of overall model performance

Table 1 presents the random 10-fold CV results of our proposed method, showing that our MDA8 estimates closely align with the measured MDA8 $O_3$ concentrations. For the CV over the entire modeling period, the R² values for daily and monthly MDA8 estimates were 0.83 and 0.96, respectively. The corresponding RMSE (MAE) values were 18.89 μg/m³ (13.71 μg/m³) for daily estimates and 7.15 μg/m³ (5.12 μg/m³) for monthly estimates. At the provincial level, our XGBoost model excelled in Beijing, Tianjin, Hebei, Shanxi, and Henan provinces/cities, achieving CV R² values above 0.86, but it performed less well in Fujian and Taiwan, where R² values were below 0.70 (Fig. S3). When examined by year, the R² [RMSE] values improved from 0.76 [24.03 μg/m³] in 2014 to 0.87 [14.67 μg/m³] in 2020. This improvement is primarily attributed to the increased sample size resulting from

the addition of more monitoring stations in later years. Additionally, the estimation accuracy metrics at the monthly level were significantly better than those at the daily level, suggesting that temporal averaging can mitigate the uncertainty in model estimates. Overall, focusing on estimation accuracy, our proposed method achieves performance that is superior to, or at least comparable with, previous ozone modeling studies, with sample-based 10-fold CV R² values at the daily level ranging from 0.70 to 0.87 (Table S4) (Chen et al. 2021; Liu et al. 2020; Ma et al. 2022a; Wei et al. 2022; Xue et al. 2020; Zhu et al. 2022).

Table 1. The random 10-fold CV results for the proposed long-term MDA8 $O_3$ modeling method.

| Period | Daily | Monthly |
| --- | --- | --- |





| | Sample size | $R^2$ | Slope | Intercept | RMSE ($\mu g/m^3$) | MAE ($\mu g/m^3$) | $R^2$ | Slope | Intercept | RMSE ($\mu g/m^3$) | MAE ($\mu g/m^3$) |
|---|---|---|---|---|---|---|---|---|---|---|---|
| All | 3,249,653 | 0.83 | 0.81 | 17.74 | 18.89 | 13.71 | 0.96 | 0.91 | 8.22 | 7.15 | 5.12 |
| 2014 | 292,643 | 0.76 | 0.75 | 23.70 | 24.03 | 17.00 | 0.94 | 0.88 | 12.43 | 9.85 | 7.08 |
| 2015 | 476,631 | 0.78 | 0.77 | 21.20 | 21.35 | 15.51 | 0.94 | 0.88 | 11.70 | 9.01 | 6.65 |
| 2016 | 466,618 | 0.79 | 0.78 | 20.25 | 20.37 | 14.92 | 0.95 | 0.89 | 9.92 | 8.14 | 6.03 |
| 2017 | 498,439 | 0.84 | 0.83 | 16.92 | 18.61 | 13.76 | 0.96 | 0.92 | 6.96 | 6.83 | 5.10 |
| 2018 | 496,152 | 0.84 | 0.82 | 16.78 | 17.89 | 13.24 | 0.97 | 0.93 | 6.43 | 6.07 | 4.59 |
| 2019 | 498,544 | 0.87 | 0.86 | 13.63 | 16.56 | 12.30 | 0.98 | 0.95 | 4.85 | 5.36 | 3.95 |
| 2020 | 520,626 | 0.87 | 0.87 | 12.59 | 14.67 | 10.86 | 0.98 | 0.96 | 4.13 | 4.28 | 3.19 |

### 3.1.2 Evaluation of pre-2013 estimates

We employed a rigorous validation approach, namely the leave-one-year-out CV, to assess the model's predictive capability in
180     years lacking national ground-level $O_3$ monitoring data. Figure 1a illustrates that our proposed modeling framework predicts
historical $O_3$ data with somewhat high estimation uncertainty at the daily level (i.e., $R^2$=0.57, RMSE=29.72 $\mu g/m^3$, MPE=22.11
$\mu g/m^3$) and reduced uncertainties at the monthly level (i.e., $R^2$=0.74 and RMSE=17.75 $\mu g/m^3$, MPE=12.76 $\mu g/m^3$). Additionally,
an independent evaluation using ozone measurements from Hong Kong demonstrates that our model achieved $R^2$ and RMSE values
of 0.41 and 41.95 $\mu g/m^3$ (Fig. 1b), respectively. These results are comparable to those from the leave-one-year-out CV conducted
over Hong Kong (i.e., $R^2$ of 0.44, RMSE of 32.84 $\mu g/m^3$, MPE=24.86 $\mu g/m^3$ in Table S5). This consistency underscores that the
reliability of the leave-one-year-out CV for assessing model's predictive performance in periods without national ground-level
ozone measurements.



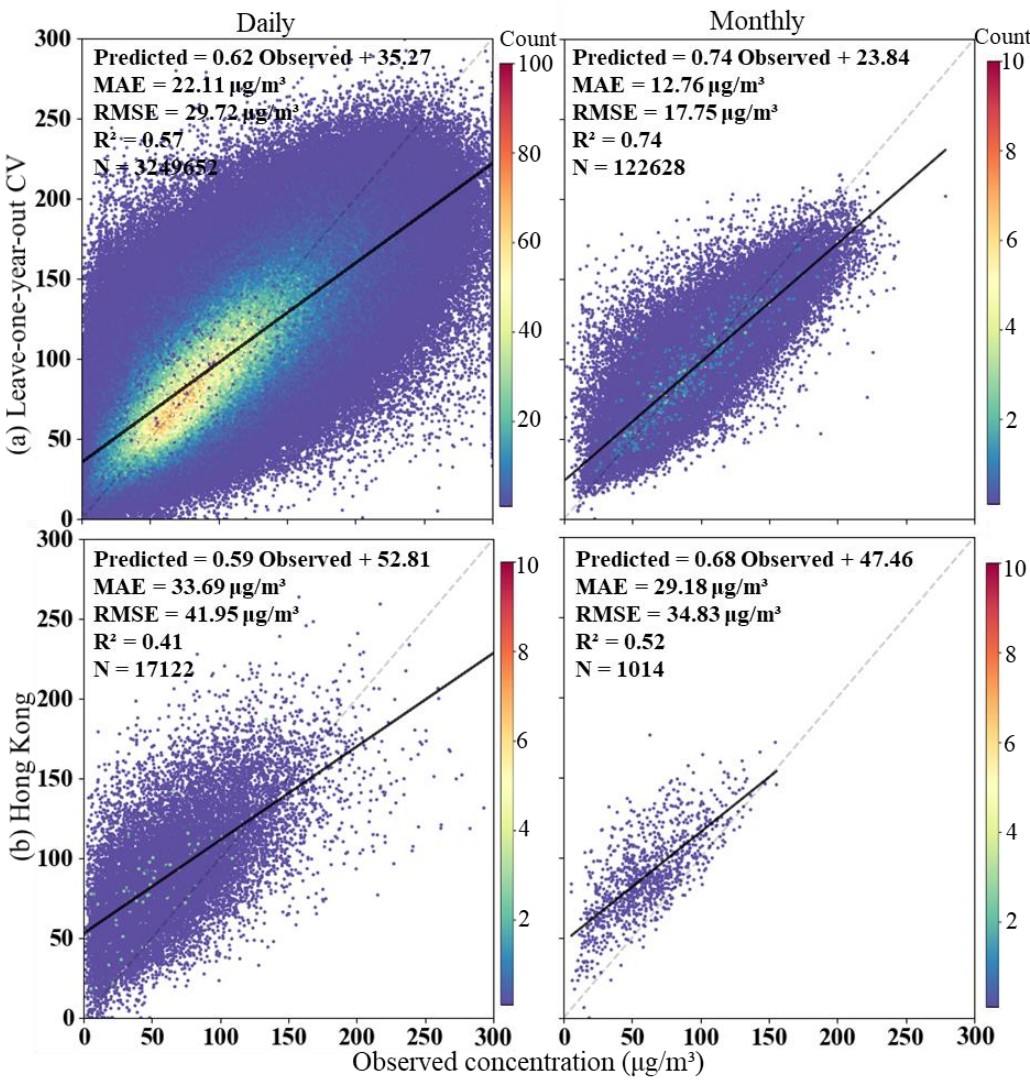

Figure 1. Validation results of historical MDA8 O₃ estimates: (a) leave-one-year-out CV at daily and monthly levels; (b) independent validation results against monitoring data from Hong Kong from 2005 to 2012, where the monitoring data over Hong Kong have not been employed in model development.

While several studies have developed long-term O₃ estimation models for China, few have quantitatively evaluated the predictive accuracy of their pre-2013 estimates (Table S4). Liu et al. (2020) and Ma et al. (2022a) reported that these models predicted pre-2013 MDA8 O₃ concentrations with leave-one-year-out CV $R^2$ [RMSE] of 0.69 [19.47 µg/m³] and day-based 10-fold CV $R^2$ of 0.63 at the monthly level, respectively. Compared with these earlier long-term ozone modeling studies (Table S4), our results demonstrate more reliable for historical years without ground-level ozone measurements, with stronger leave-one-year-out CV and day-based 10-fold CV results, particularly at the aggregated monthly level (Chen et al. 2021; Liu et al. 2020; Ma et al. 2022a; Wei et al. 2022; Xue et al. 2020; Zhu et al. 2022). These findings indicate that our model not only captures long-term trends but also



the fine-scale variations in ozone across China more accurately, making it valuable for both long-term and short-term ozone variation and exposure research.

### 3.1.3 Estimated high-resolution maps of ground-level ozone

We selected a subset from June 2018 of our long-term, full-coverage MDA8 O$_3$ estimates generated by our proposed modeling framework to evaluate whether the extrapolated surfaces accurately represent the day-to-day and fine-scale variations in ground-level ozone concentrations. Figure 2 (a-d) presents a comparison of monthly mean ground-level MDA8 O$_3$ concentrations from our high-resolution estimates with the 10-km MDA8 estimates by Wei et al. (2022) and in-situ measurements for June 2018 — a month noted for high ozone concentrations at the monitoring stations. The nationwide distributions illustrate that our model

successfully captures the general spatial variation pattern of ground-level ozone across China, aligning well with both Wei et al. (2022)'s findings and measured values (Fig. 2a). Zoom-in maps of Jinan, Wuhan, and Chongqing highlight that our modeling approach predicts fine-structures in ground-level ozone concentrations that are not discernible in coarser-resolution maps and in-situ measurements (Fig. 2 b-d). Additionally, comparisons of daily time series of MDA8 O$_3$ estimates versus observations in 2018 (Fig. 2d & Fig. S4) show that our method effectively captures daily and seasonal variability, although it tends to underestimate

extremely high concentrations. This underestimation is likely due to the regression approach, which optimizes predictions based on average behavior. Overall, while 10-km and in-situ data primarily identify broad 'hotspots' of ground-level ozone (e.g., at the city scale), our high-resolution predictions uncover much more intricate structures, capturing sharp spatial and temporal gradients shaped by both natural and anthropogenic factors.

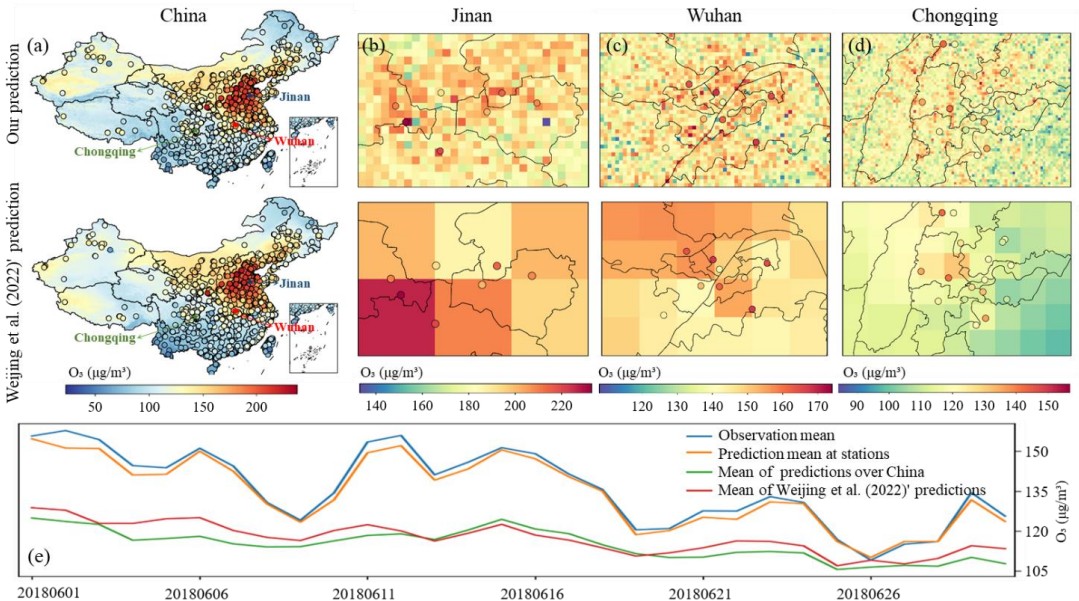



Figure 2. Spatiotemporal comparisons of ground-level ozone predictions vs. observations in China in June 2018: (a) monthly mean map of MDA8 O₃ predictions with station ozone measurements (monthly valid measurements for each station >15); zoom-in maps over (b) Jinan, (c) Wuhan, and (d) Chongqing; (e) Time series of mean MDA8 O₃ concentrations over China during June 2018.

## 3.2 Spatial distribution and long-term trend of ground-level O₃ exposure

Figure 3a illustrates how long-term pollution hotspots vary by region and season. In spring, moderate O₃ pollution widespread in the eastern region, with most MDA8 O₃ concentrations ranging between 110 μg/m³ and 130 μg/m³, except in the southern provinces. Summer sees severe ozone pollution, with concentrations exceeding 100 μg/m³ in most areas, apart from the Qinghai-Tibet plateau and Yunnan province. During this season, the central North China Plain (NCP) experiences the highest polluted levels, with 21-year mean MDA8 O₃ concentrations routinely surpassing 140 μg/m³. In autumn, the southern provinces experience mild ozone pollution with most MDA8 O₃ levels between 100 μg/m³ and 110 μg/m³, while the Pearl River Delta (PRD) becomes the prominent ozone exposure hotspot, with most concentrations exceeding 120 μg/m³. Winter features the lowest ozone levels, with nearly all regions recording MDA8 O₃ concentrations below 100 μg/m³. Consequently, we identified eastern China, the NCP), and the PRD as long-term ozone high-exposure regions, which were given special attention in the subsequent analysis.

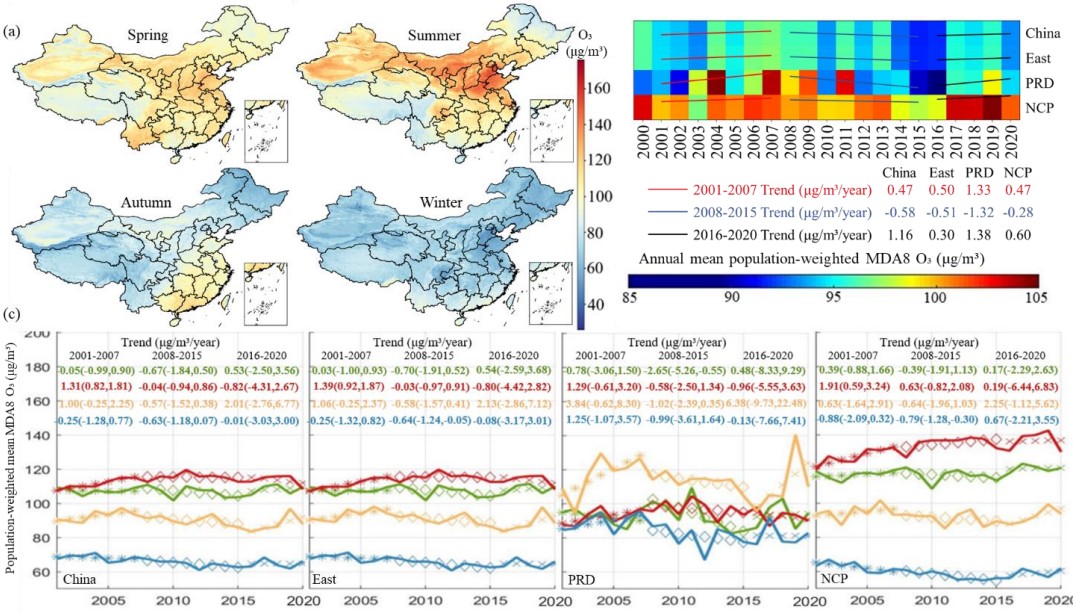

Figure 3. Overall spatiotemporal patterns of exposure to ground-level ozone in China over the past two decades: (a) Spatial map of seasonal mean MDA8 O₃ concentrations during 2000-2020; (b) annual mean population-weighted MDA8 O₃ concentrations over China and the three regions (i.e., eastern China, PRD, and NCP); (c) Seasonal population-weighted mean MDA8 O₃ concentrations over China and three typical regions, and their linear trends (green for spring, red for summer, orange for autumn, and blue for winter), where asterisks (*), diamonds (◊), and crosses (x) correspond to the linear trends in the 2001–2007, 2008–2015, and 2016–2020 sub-periods, with 95% confidence intervals and colors matched to seasons.





By integrating yearly population distributions from Landscan (Rose et al. 2020), we analyzed the spatiotemporal patterns of exposure to ground-level ozone across China over the past two decades. Figure 3 (b-c) shows the annual and seasonal trends of population-weighted mean MDA8 $O_3$ concentrations, revealing long-term trends that are non-monotonous and vary significantly across different regions. As illustrated in Fig. 3b, national level trends feature two pivotal changes around 2008 and 2015: in the first phase of 2001-2007, the population-weighted exposure to ozone increased with a linear slope of 0.47 μg/m$^3$/year, then decreased post-2008 with a slope of -0.58 μg/m$^3$/year, followed by a substantial rise at a rate of 1.16 μg/m$^3$/year during the third phase of 2016-2020. These shifting exposure trajectories also displayed pronounced seasonal and regional variations (Fig. 3c). During the 2001-2007 phase, eastern China and the NCP experienced a significant rise in ozone exposure during the summer months, with slopes ranging from 1.39 to 1.91 μg/m$^3$/year, whereas the PRD region saw its most substantial increases in autumn, with a notable slope of 3.84 μg/m$^3$/year. In the second phase, the three typical ozone hotpots generally showed decreasing trends across all seasons (slopes from -0.03 to -1.02 μg/m$^3$/year), except for a slight increase during the summer in the NCP (slope of 0.64 μg/m$^3$/year). Moving into the 2016-2020 phase, these typical hotspots transitioned to marked increasing trends in autumn, particularly in the PRD, which displayed a steep increase with a slope of 6.38 μg/m$^3$/year. Conversely, while other regions exhibited declining trends during the summer season of this phase, the NCP continued to show an upward trend with a slope of 0.19 μg/m$^3$/year.

### 3.3 Monthly exposure and county-level pattern

To investigate how the most severe ozone pollution events and baseline levels have evolved over time, we further analyzed monthly exposure patterns, focusing specifically on the trends in monthly population-weighted mean MDA8 ozone concentration peaks and troughs across China and three key regions. Overall, the monthly ozone concentrations followed a three-phase trend, similar to the annual patterns identified earlier (as shown in Fig. S5 and Fig. 3b), with slight regional variations in the slopes. However, a closer examination of the monthly peaks and troughs revealed distinct changes. As indicated in Fig. 4a, all regions experienced an increase during the first phase, with the PRD recording a notable rise at a rate of 4.37 μg/m$^3$/year. The second phase showed general declines across most regions (slopes range from -0.99 to -1.19 μg/m$^3$/year), except for the PRD (slope=-0.009 μg/m$^3$/year), which remained relatively stable. The subsequent phase again saw increases in all regions, ranging from 0.19 to 0.44 μg/m$^3$/year, with the PRD experiencing a significant uptick at a slope of 3.19 μg/m$^3$/year. The trends in monthly troughs mirrored the decline observed in the second phase, albeit with varied patterns in other stages. From 2001 to 2007, both China and eastern China registered slight increases, whereas the NCP and PRD noted decreases in trough levels. During the 2016-2020 period, the PRD showed a marked increase, with a slope of 2.81 μg/m$^3$/year, contrasting with decreases ranging from -0.33 to -0.52 μg/m$^3$/year in the other regions. Additionally, over the past two decades, the monthly peaks predominantly occurred in June. However, results from the Mann-Kendall test (Table S6) indicate a significant shift in the timing of peak ozone concentrations across most of China, with p-values below 0.05 for China, the east, and the NCP, suggesting a potential shift from June to earlier in May in recent years.



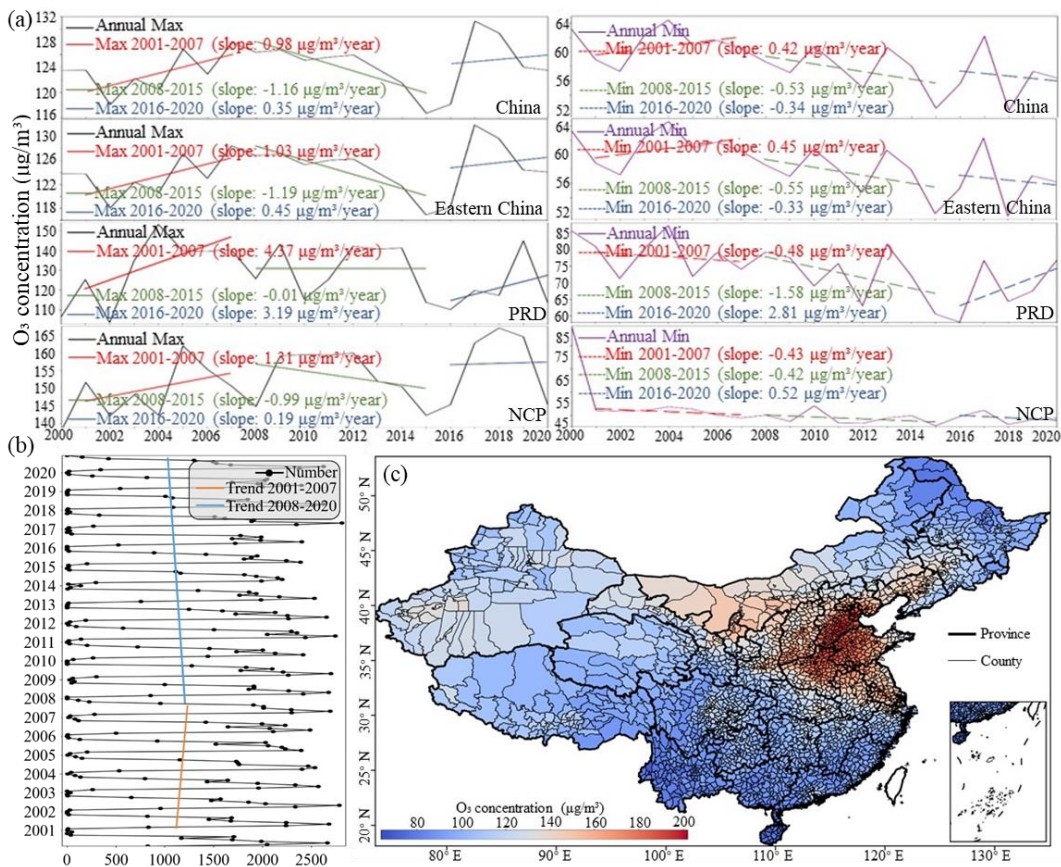

Figure 4. Spatiotemporal distributions of monthly population-weighted mean MDA8 O₃ concentrations over China and the three typical regions: (a) time series with three-piece trend lines of peaks (left panel) and troughs (right panel) from 2001 to 2020; (b) spatial distribution of county-level mean concentrations in June 2018.

To provide a spatial and temporal overview of severe ozone pollution trends across counties in China, we tracked the number of counties where monthly ozone concentrations exceeded 100 μg/m³ from 2000 to 2020. As shown in Fig. 4b, the trend exhibits significant seasonal variation each year, with peaks typically occurring in late spring and early fall. While the seasonal pattern remains consistent annually, the amplitude of these peaks fluctuates from year to year. Notably, in May 2017, 2,812 counties exceeded the ozone threshold, compared to an average of 2,543 counties during the month of May in other years. On average, 1,142 of the 2,900 counties exceeded this 100 μg/m³ threshold monthly, with the 25th and 75th percentiles at 38 and 2,002 counties, respectively. Interestingly, the time series for the number of counties with exceedances did not exhibit the three-phase variation identified earlier. Instead, the number of counties exceeding the threshold increased from 2001 to ~2007, with a slope of 16.84 per year, and then decreased in the subsequent years, with a slope of -12.44 per year.





Figure 4c displays a county-level spatial map from June 2018, a month identified as a hotspot for high ozone levels, highlighting
significant spatial disparities in ozone exposure within cities. For example, in Beijing, the population-weighted mean MDA8 O₃
concentrations ranged from 141.23 μg/m³ in Yanqing in the northwest to 180.33 μg/m³ in Tongzhou in the southeast. Nationally,
the highest exposure levels were recorded in Xiqing and Beichen counties in Tianjin, with concentrations around 200 μg/m³.
Conversely, the lowest exposures were observed in two southwestern counties in Yunnan province and three southeastern counties
in Hainan province, with concentrations below 70 μg/m³.

**3.3 Short-term exposure characteristics and extreme episodes of ground-level O₃**

We observed the day-to-day variation in ground-level ozone concentrations for China as a whole and for the three typical regions
over the past two decades. The coefficient of variation of the daily MDA8 O₃ predictions indicates significant spatial heterogeneity
in the nationwide distribution of ambient ozone, with values ranging from 0.16 to 0.41 (Fig. S6). This variability is characterized
by notable seasonality, displaying the highest mean values in autumn (0.29) and the lowest in spring (0.22).

Overlaying daily MDA8 O₃ predictions with population distribution, Fig. 5a reveals that from March 2000 to December 2020,
over 60% of the Chinese population was exposed to MDA8 O₃ concentrations exceeding 100 μg/m³, defined as the first level of
the national ambient air quality standard, on more than 31% of the total prediction days. The long-term variation of these exposure
ratios follows a three-piece pattern (Fig. S7) similar to the annual ozone exposure trend identified in Section 3.2. The highest
exposure months are May and June, with daily ratios around 70%. Particularly in May 2007 and 2017, the average ratio reached
79%, with ranging between 56% and 97%. At the regional level, daily ozone exposure in the three typical ozone exposure hotspots
was more severe than the national average, especially for the NCP and PRD regions. In these regions, ~60% and ~40% of days,
respectively, saw the population exposed to ozone levels above 100 μg/m³. Furthermore, ~5% of days in the NCP and ~2% in the
PRD exceeded the national second-level limit of 160 μg/m³ (Fig. 5b). The spatial map in Fig. 5c further illustrates that extremely
severe ozone exposure was concentrated in the NCP region, especially for the central NCP, with most area experiencing more than
10% of days with concentration exceeding 160 μg/m³. Over time, this severe exposure expanded from part of south Hebei, Tianjin,
north Henan, and west Shandong in the first phase of 2001-2007 to cover most of the NCP region in the third phase of 2016-2020.



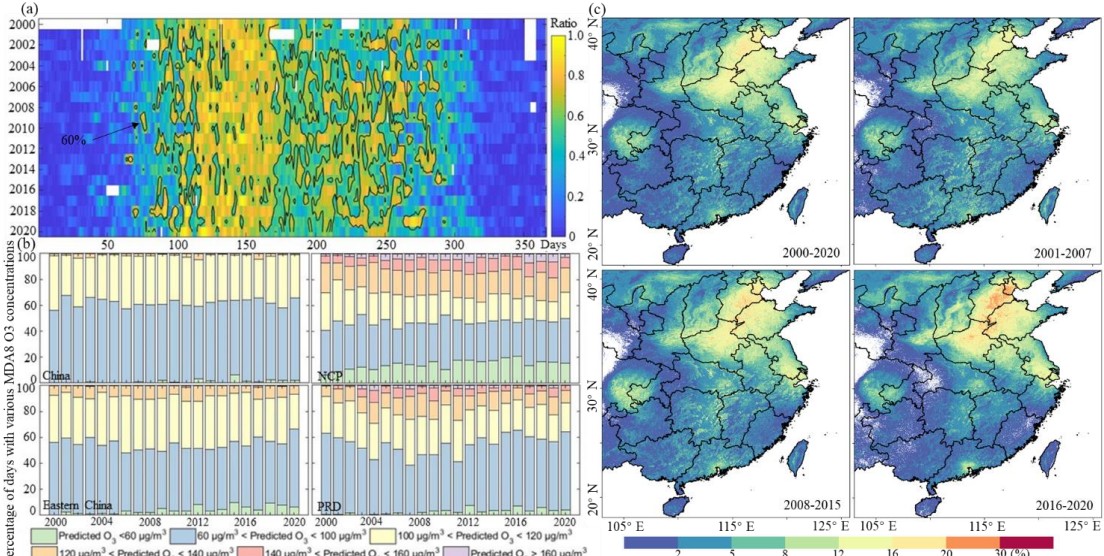

Figure 5. Day-to-day patterns of ground-level ozone exposure levels from 2000 to 2020: (a) heatmap of daily ratios of the
population exposed to MDA8 $O_3$ concentration exceeding 100 µg/m³; (b) percentage of days with various MDA8 $O_3$ concentrations;
and (c) high-exposure risk maps calculated for 2000-2020 and the three phases.

The severe ozone pollution events are usually associated with meteorological conditions (Yang et al. 2024). Figure 6 exemplified
an extreme ozone pollution episode on 25 June-5 July of 2017 over the NCP. The West Pacific Subtropical High, positioned
between 20–26°N, significantly influenced ozone distribution over the Yangtze River Delta by modulating precipitation and solar
radiation. High relative humidity in the region suppressed ozone formation in this region, resulting in lower concentrations. In
contrast, the Beijing-Tianjin-Hebei and its surrounding area (BTH) experienced favorable conditions for ozone production under
an anomalous high-pressure system in the upper troposphere (Xu et al. 2019). Warm, southerly winds in the lower troposphere
contributed to higher temperatures and northward transport of aged air masses, increasing ozone and its precursors. Furthermore,
a persistent temperature inversion in the BTH trapped pollutants in lower layers, exacerbating ozone pollution. This inversion
preserved ozone at night and facilitated its descent to the surface at sunrise, worsening the pollution. These persistent conditions
sustained severe regional ozone pollution events in the BTH (Mao et al. 2020).



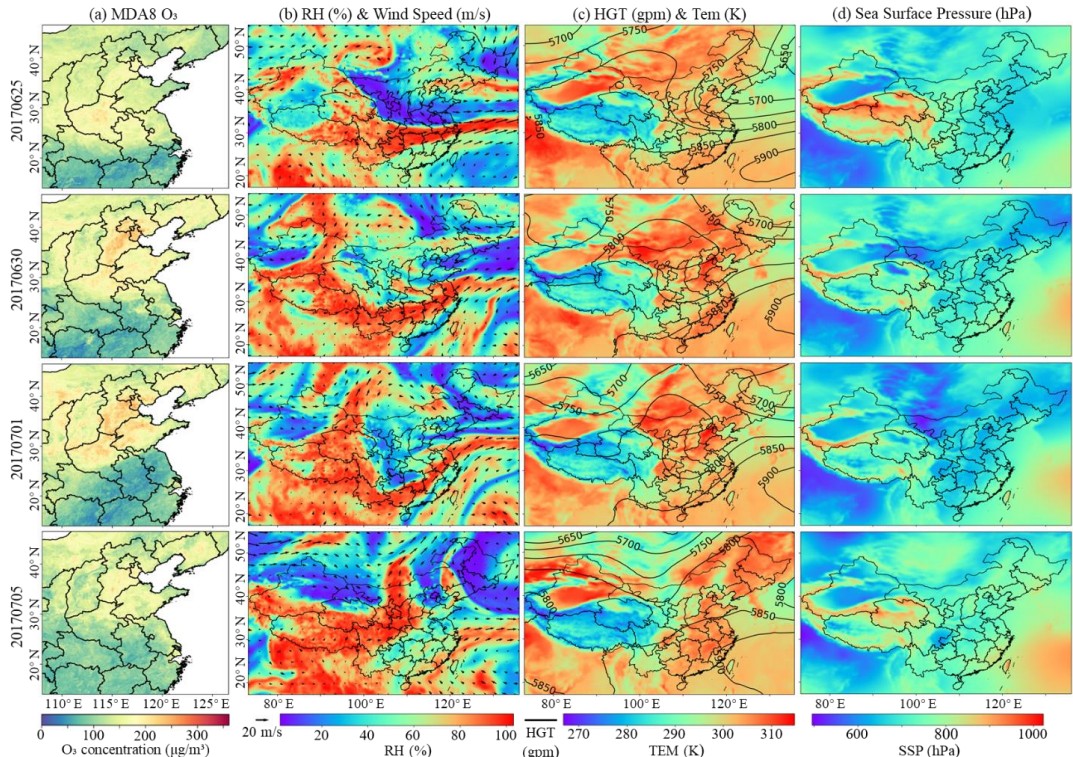

Figure 6. Spatial distributions of daily MDA8 O₃ and meteorological fields (relative humidity (RH), wind speed (WS) at 500 hPa,
geopotential height field (HGT) at 500 hPa, air temperature (T) at 2m, and sea surface pressure (SSP)) in 25 June -5 July 2017.

## 4 Discussion

In this study, we developed an XGBoost-based prediction model to hindcast long-term, full-coverage ground-level MDA8 $O_3$
concentrations across China with daily and 1-km spatiotemporal resolution. To the best of our knowledge, our model's performance,
with a sample-based 10-fold CV $R^2$ of 0.83 at the daily level, is comparable to other national-scale ozone modeling studies in China,
which have reported validation $R^2$ values ranging from 0.77 to 0.87 (Table S4). Despite similar performance levels, our modeling
framework offers superior spatiotemporal resolution (daily and 1-km vs. monthly or 0.05° or coarser) and covers a longer prediction
period (2000-2020 vs. 2005-2019 or shorter) compared to these studies. The rigorous hindcast and individual validation results
confirm that our long-term estimates reasonably represent day-to-day trends and intra-urban variations in ground-level ozone,
including for the pre-2013 period. The long-term, full-coverage estimates capture short-term local pollution variations and details
not revealed by previously coarser resolution or shorter-period data (Fig. 1 and 3). For example, Fig. S8 illustrates a case study
from Wuhan on May 28 2017, a day characterized by elevated ozone levels, showing local $NO_2$ levels titrated $O_3$, resulting in
observed ozone concentrations that are lower than those downwind. Consistent with our previous findings (He et al. 2024), the
incorporation of LST, which is closely related to ozone variations and available at high spatiotemporal resolutions, significantly
enhances the quality of the estimation data (Fig. S9). Additionally, the inclusion of other spatiotemporal covariates related to ozone



formation and dispersion, such as radiation, terrain, and the ozone precursor NO$_2$, helps the model capture the complex spatiotemporal dynamics of ground-level ozone.

Our long-term trend analysis reveals a three-phase variation pattern in ground-level ozone across China over the past two decades, characterized by significant seasonality and region disparities (Fig. 3, Fig. S5, fig. 4). This pattern likely results from a complex interplay of environmental, regulatory, and climatic factors influencing ozone levels. Similar to long-term PM$_{2.5}$ in China (He et al. 2023), the first 2001-2007 phase presented an increasing trend, possibly linked to rapid industrial growth and urbanization accompanied by lenient environmental controls. This period likely saw higher emissions of ozone precursors such as volatile organic compounds (VOCs) and nitrogen oxides (NO$_x$) (Ahammed et al. 2006; Akimoto 2003). The second phase featured a general decline in ozone levels, probably attributed to stricter air quality policies implemented by the central government, including notable reductions in nitrate emissions observed during 2012-2016 (Wang et al. 2019). Despite these improvements, the summer increase in the NCP suggests that some sources of ozone precursors were not fully addressed by the existing measures. More crucially, despite the implementation of even stricter air pollution controls in the third phase, the declining trends in national and eastern ozone levels significantly slowed down, accompanied by a shift toward overall increasing trends in the PRD and NCP regions (Fig. S5 and Fig. 3). This suggests that while the initial measures were effective, they may require adjustments to cope with new challenges, such as increased vehicular traffic, ongoing industrial activities, and climatic changes conducive to ozone formation (Li et al. 2024a; Ruosteenoja and Jylhä 2023; Wang et al. 2019). Moreover, the differing trends, predominantly increasing in summer in northern areas and in autumn in southern areas, underscore the influence of regional weather patterns on ozone dynamics. Therefore, policies should be specifically tailored not only to distinct regions but also to particular seasons where ozone peaks are most significant.

The distribution of ozone exposure hotspots presents significant seasonal changes, with summertime hotspots predominantly occurring in the NCP and shifting to the PRD during autumn (Fig. 3). The most severe ozone exposures, with concentrations exceeding 160 µg/m³, were typically observed in the NCP. This region has experienced an increasing trend in both the geographical extent and frequency of these high concentrations (Fig. 5). Temporally, we observed a notable shift in the peak ozone exposure month from June to May, especially pronounced in the NCP. This escalation in ozone pollution levels and the earlier annual peak may be attributed to changes in meteorological conditions, such as extreme high temperatures (Wang et al. 2022a), and air pollutant emissions, notably the reduction in NO$_x$ emissions coupled with high emissions of VOCs (Ke et al. 2021), which are conducive to ozone formation. Furthermore, previous studies have suggested that a decrease in organic and black carbon aerosols may elevate daytime MDA8 O$_3$ levels (Li and Li 2023). Thus, the significant reduction in ambient particulate matters in the NCP in recent years has also contributed to worsening ozone conditions in this region. Consequently, these shifts in spatial and temporal exposure hotspots should raise concern among government officials, as the expansion in both the extent and duration of high ozone levels could lead to more severe public health impacts and damage to agricultural production.

**5 Conclusion**

In this study, we developed a comprehensive, high-resolution modeling method using the XGBoost algorithm to hindcast ground-level ozone concentrations across China. Utilizing a combination of ground-level ozone observations, high-resolution LST, other



atmospheric parameters, and geographical data, our model achieved notable prediction accuracy and maintained fine spatiotemporal resolution, full coverage, and an extended temporal range, covering 13 years prior to the establishment of the national air quality monitoring network. Rigorous validations using leave-one-year CV and external ground-level ozone measurements show that our pre-2013 estimates accurately represent long-term ozone variations, and spatial map comparisons demonstrate that our high-resolution estimates capture more short-term local variations than previous datasets. Utilizing the proposed modeling method, we generated daily ozone estimates from 2000 to 2020, exploring variations and exposure characteristics across multiple spatial and temporal scales and gained findings as below. These findings underscore the urgency of addressing the long-term trends and recent changes in ozone variation for public health and suggest that further actions are required to mitigate these effects. Our results provide essential data for environmental policy-making and health studies, offering a basis for targeted interventions aimed at reducing ozone exposure risks.

(1) **Seasonal hotspot shifts and non-monotonous trends**: High-level ozone centers shifted seasonally, with spring concentrations primarily in eastern China, summer in the NCP, and autumn in the PRD. A three-phase, non-monotonous long-term trend was observed: an increase in the first phase (2001-2007) with a slope of 0.47 μg/m³/year, a decrease post-2008 with a slope of -0.58 μg/m³/year, and a significant rise during 2016-2020 with a slope of 1.16 μg/m³/year. This trend was marked by significant regional and seasonal fluctuations. Notably, the NCP experienced an increasing trend in summer during the second phase (slope of 0.63 μg/m³/year), whereas other regions generally showed seasonal declines. Importantly, the PRD exhibited a significant upward trend in autumn during the third phase, with a slope of 6.38 μg/m³/year.

(2) **Increased exposure risk recently**: Risks of ozone exposure, with concentrations exceeding 100 μg/m³, typically peaked in June but have recently shifted earlier to May. Moreover, dangerous exposure levels exceeding 160 μg/m³ were concentrated in the NCP, with a trend of expanding in recent years. Our day-to-day analysis of ozone pollution episodes highlights the significant impact of meteorological conditions on extreme ozone events.

**Data availability**. The final ozone estimates will be available at https://doi.org/10.5281/zenodo.4569557.

**Author Contributions**. Q. He designed the study framework, and conducted the formal analysis. J. Cao, T. Ye and W. Wang processed the data, developed the code and validated the estimation model and visualized the results. The manuscript was initially written by Q. He and revised by Pablo E. Saide.

**Competing interests**. At least one of the (co-)authors is a member of the editorial board of Atmospheric Chemistry and Physics.

**Finial support**. This study is supported by the National Natural Science Foundation of China (Grant NO. 41901324).

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
