# Peer review of "Text S1. Data and Methodology for long-term ozone modeling"

_EGUsphere, 2024_

## Author Comment (AC1)

**Response to Reviewer 1:**

*Comments:*

*He et al. developed a machine learning model to estimate gridded ozone data using meteorological parameters, pollutant variables, geographical covariates, temporal dummy variables, and ground station observation data. They conducted the modeling based on data after 2014 and extended the ozone estimates to the period 2000-2013, ultimately generating seamless ozone data for 2000-2020. Detailed exposure risk analyses were also conducted. Given the persistently high ozone concentrations in China in recent years, the data and analysis results presented in this paper are of value for decision-making. Overall, the paper generates long-term ozone data, the methods and framework employed are reasonable, and the results and analysis offer some insightful takeaways, but there are still several issues that need to be addressed:*

   **Response**: Thank you for reviewing our manuscript and providing us constructive suggestions/comments. We have thoroughly considered your suggestions/comments and revised our manuscript accordingly. The item-by-item responses are below.

*(1).The authors emphasize the importance and contribution of surface temperature as a proxy. However, two aspects warrant further exploration:*

*(a) The conclusion that surface temperature is important is derived from the model's variable importance analysis. What would happen if surface temperature were removed from the model? Would the overall modeling accuracy decrease, and by how much? Additionally, how would the spatial mapping results be affected in the absence of high-resolution surface temperature data? A visual example to illustrate this would be helpful.*

   **Response**: Thank you for your valuable suggestion. We have conducted a series of experiments to evaluate the impact of incorporating the satellite-derived land surface temperature (LST) variable into ground-level ozone models on estimation accuracy. Compared to the baseline model, the inclusion of LST significantly improves predictive performance, with $R^2$ values increasing by 0.04–0.06 across random, site-based, and day-based 10-fold cross-validation (CV). These findings have been incorporated into the first paragraph of Section 4 in the revised manuscript as shown below.

   "…Consistent with our previous findings (He et al., 2024), incorporating LST—closely linked to ozone variations and available at high spatiotemporal resolutions—significantly enhances the overall quality of our estimation data. This improvement is demonstrated by its leading rank in variable importance (Fig. S9) and the observed increase in $R^2$ values by

0.04–0.06 across sample-, site-, and day-based 10-fold CVs when comparing models with and without LST (Table S8). …"

Additionally, we examined the effect of satellite-derived LST on the spatiotemporal mapping of ground-level ozone using an interpretable machine-learning approach. The modeling process and key findings, detailed in our previous publication (He et al., 2024), are summarized below. Since the modeling framework in this study builds upon our previous work, and the primary objectives of this study are to evaluate spatial hotspots of ground-level ozone and their temporal changes, we have not redundantly discussed the role of satellite LST in mapping in the present manuscript.

> ➢ **Variable Importance Analysis.** We applied both global (impurity-based) and localized (SHapley Additive exPlanations, SHAP) variable importance methods to analyze the contributions of each predictor in the optimally trained model. Satellite-derived LST consistently ranked as the most important variable, contributing approximately 32% (impurity-based) and 22% (SHAP) to the model's predictive performance. Analyses at varying spatial resolutions (1 km, 10 km, and 25 km) further underscored the critical role of satellite LST in high-resolution ozone mapping, potentially surpassing the influence of other meteorological factors.

> ➢ **Comparison with Alternative Temperature Data.** We investigated the impact of replacing LST with reanalyzed 2-meter air temperature (T2M) data. The results revealed significant shifts in variable importance rankings. The temporally dummy variable (TX) became the most important predictor (~15%), with T2M ranking second (~12%) in the localized SHAP analysis. When we developed a 25-km resolution ozone prediction model, T2M maintained a second-place ranking (~11%), but its importance fell short of expectations. Even at comparable spatial resolutions, T2M captured less predictive information than LST, suggesting that satellite-derived LST's combined spatial and temporal high resolution provides superior information for ozone modeling.

> ➢ **Robustness Across LST Sources.** To test the robustness of LST's role, we replaced TPDC-LST with MODIS LST in the ozone prediction model. LST consistently emerged as the most critical variable, contributing approximately 40% in the impurity-based importance analysis. The MODIS LST-based model maintained strong performance, achieving a random 10-fold CV $R^2$ of 0.91 and an RMSE of 14.16 μg/m³. These results reaffirm the critical role of LST in ozone prediction, irrespective of the satellite data source, and highlight the robustness of LST in capturing key ozone-relevant information.

*(b) We are aware of global warming. If surface temperature plays a decisive role in ozone estimation, this may be suitable for the integration between ozone and surface temperature*

*observation, but is it equally applicable for hindcasting? Further discussion on this point could enhance the scientific value of the paper.*

**Response**: Thank you for highlighting the effect of LST on ozone hindcasting, which is an excellent point for further discussion. To address this, we have developed models with and without LST as a predictor and compared their CV results to demonstrate the contribution of LST to ozone hindcasting. These results have been incorporated into the first paragraph of Section 4, as shown below.

"… Additionally, its critical role in hindcasting ground-level ozone estimates for the pre-2013 unmonitored period is validated through improvements in estimation accuracy, as reflected by an $R^2$ increase of 0.07 in the leave-one-year-out CV (Table S8) and 0.02 in independent validation using Hong Kong in-situ measurements (Table S9). …"

*(2). The paper estimates historical ozone data over an extended period, which is difficult to validate. While observation data from Hong Kong are scarce, they provide valuable validation. However, this is not clearly explained in the paper. How many observation sites in Hong Kong were used, and what time periods do the data cover? This information is crucial for assessing the accuracy of the historical ozone estimates. A more in-depth analysis would be beneficial. For example, if a few years of data are available, how does the accuracy vary year by year?*

**Response**: We appreciate your comment and fully agree that detailed information about the Hong Kong observational data is crucial for assessing the accuracy of historical ozone estimates. In our study, the independent validation dataset from Hong Kong consisted of 17,122 observations spanning the period from 2005 to 2012. Please note that 2005 is the earliest year the Hong Kong measurements are available. A statistical summary of the model's performance across different years is presented in Table S5 in the revised version, and the relevant content has been updated as shown below.

"…Additionally, an independent evaluation using 17,122 ozone measurements from Hong Kong, spanning 2005 to 2012, demonstrates that our model achieved $R^2$ values ranging from 0.31 to 0.59 and RMSE values from 34.65 to 45.40 $\mu g/m^3$, with averages of 0.41 and 41.95 $\mu g/m^3$, respectively (Fig. 1b and Table S5)."

*(3). The time periods are divided into 2001-2007, 2008-2015, and 2016-2020 for trend analysis. Were these periods defined based on trend identification, or was the segmentation arbitrary? This needs to be clarified in the paper.*

**Response**: The three periods were defined based on the national annual exposure time series shown in Fig. 3b. We observed two turning points, around 2008 and 2015, in the annual mean population-weighted MDA8 $O_3$. The relevant sentence has been revised for

clarity, as shown below:

"…As illustrated in the annual exposure time series (Fig. 3b), two turning points are observed around 2008 and 2015…"

(4). The paper presents results across multiple scales (e.g., pixel, county, region, national), offering a comprehensive view of the analysis from different perspectives. Figure 4 presents fine-grained county-level analysis, but it is unclear what new insights this scale of analysis provides. It seems neither as detailed as pixel-level analysis nor as regionally distinctive as the regional-scale analysis. It would be helpful to clearly state the key conclusions derived from this level of analysis.

**Response**: In the revised manuscript, we have clearly articulated the key conclusions of the county-level analysis, highlighting its unique contributions. In general, the county-level analysis is particularly valuable for effective environmental management, as it enables tailored interventions in areas with elevated ozone concentrations. The following content has been incorporated into the penultimate paragraph of Section 4 in the revised manuscript.

"Based on the high-resolution estimates, we quantitatively identified counties with the highest and lowest ozone levels (Fig. 4b-c), offering critical insights to inform resource allocation and targeted pollution control measures. For instance, counties such as Xiqing and Beichen in Tianjin, identified as having high ozone levels, can be prioritized for implementing targeted emission control policies and public health campaigns to mitigate health risks for local residents. These localized insights are often overlooked in broader-scale regional analyses. Previous studies relying on coarser-resolution data have typically focused on large urban agglomerations, such as the Beijing-Tianjin-Hebei region and the Pearl River Delta (PRD) (Wei et al., 2022), neglecting smaller yet critically affected areas. Conversely, while pixel-level analyses offer highly detailed spatial patterns, they may lack the administrative relevance needed for actionable policy decisions. By bridging the gap between regional and pixel-level analyses, our county-level analysis provides actionable and geographically specific recommendations, empowering policymakers to address ozone pollution more effectively."

(5). The exposure levels for the period 2016-2020 show a significant increase in the NCP region, especially compared to 2001-2007. What is the underlying cause of this increase? Additionally, the PRD has long been considered a high-concentration ozone area, but this does not seem to be reflected in the results.

**Response**: The NCP is a notable ozone hotspot, showing an increasing trend in recent years. Additionally, we observed a significant shift in the peak ozone exposure month from June to May. These changes can be attributed to shifts in meteorological conditions, such as extreme high temperatures (Wang et al., 2022), and changes in air pollutant emissions,

particularly the reduction in NOx emissions combined with high VOC emissions (Ke et al., 2021), which favor ozone formation. Moreover, the substantial reduction in ambient particulate matter in recent years has likely exacerbated ozone conditions in this region, as $PM_{2.5}$ plays a role in slowing the removal of hydroperoxy radicals, thereby promoting ozone production (He et al., 2023; Li and Li, 2023; Li et al., 2019). These points have been integrated into the third paragraph of Section 4. Please note that in response to Reviewer #2's suggestion, we have consolidated the NCP discussion into a single paragraph—the third paragraph—in the revised manuscript.

We appreciate your observation regarding the PRD region and its representation in our results. While our discussion section emphasizes the NCP region due to its predominantly exacerbated ozone pollution issues, our results do indeed reflect that the PRD has consistently been a region of high ozone exposure. This is supported by the following three key aspects:

> **Ozone exposure trends across China over the past two decades**: The PRD region has consistently experienced elevated ozone concentrations, particularly during autumn and winter, where ozone levels are significantly higher compared to other regions (Figure 3).

> **Regional population exposure:** Between March 2000 and December 2020, approximately 40% of days in the PRD region had population exposure to ozone levels exceeding 100 µg/m³, with around 2% of days exceeding the national secondary limit of 160 µg/m³ (Figure 5). This highlights the region's long-term ozone exposure burden.

> **Long-term ground-level ozone trends in the PRD region:** Our study indicates that the increasing trends in ground-level ozone exposure in the PRD are more pronounced than in most other regions. Between 2001 and 2007, ozone concentrations in the PRD rose at an average rate of 1.33 µg/m³ per year, with a particularly sharp increase of 3.84 µg/m³ per year during autumn. In the 2016–2020 period, the upward trend became even more pronounced, especially in autumn, where ozone concentrations increased by 6.38 µg/m³ per year. These trends are considerably steeper than those observed in the NCP region.

Together, these findings confirm that the PRD remains a region of long-term, high ozone exposure and underscores the need for continued attention to ozone pollution in this area.

*(6). What are the physical or chemical mechanisms through which aerosol optical depth is used to estimate near-surface ozone? Please provide further explanation.*

**Response**: The physical and chemical mechanisms linking AOD and ground-level ozone have been incorporated into Section S1.1 of the revised supporting document. The following is a more detailed explanation.

➢ Aerosols can affect the scattering and absorption of solar radiation, thereby influencing the efficiency of photochemical reactions (Wang et al., 2019), which in turn affects the formation and destruction of ozone.

➢ Additionally, certain types of aerosols (such as black carbon) can adsorb volatile organic compounds (VOCs) in the atmosphere(Gao et al., 2018), which are important precursors for ozone formation. By influencing the concentration of VOCs, aerosols can indirectly affect ozone production.

➢ Given that AOD serves as an indicator of aerosol concentration and properties, it provides important information about the potential impact of aerosols on ground-level ozone estimation.

*(7). Lines 155-167, does "province level" refer to statistical analysis by province, rather than validation by province?*

**Response**: Yes, this term indeed refers to statistical analyses conducted by province, rather than validation by province. We have further clarified this point in the revised version to ensure that readers can clearly understand. Below is the content of our modified manuscript.

"… We further compiled the CV results by province, revealing that the XGBoost model performed exceptionally well in Beijing, Tianjin, Hebei, Shanxi, and Henan, achieving CV $R^2$ values above 0.86. However, its performance was weaker in Fujian and Taiwan, where $R^2$ values fell below 0.70 (Fig. S3). …"

*(8). Figure 2 compares the results obtained in June 2018 with those from previous studies. Why was this particular time chosen for comparison?*

**Response**: We chose June 2018 as the time period for comparison primarily because this month recorded relatively high ozone concentrations in the monitoring station data, which has been identified as a hotspot for high ozone levels, highlighting significant spatial disparities in ozone exposure within cities. We have revised the opening sentence of Section 3.1.3 to clarify the reason behind selecting this subset for comparison and the relevant sentence is shown below.

"We selected a subset from June 2018, identified as a hotspot for high ozone levels based on ground-level monitoring data, from our long-term, full-coverage MDA8 O₃ estimates generated by the proposed modeling framework…"

*(9). Line 231, a writing mistake, "NCP)" should be "NCP."*

**Response**: Thank you for your careful review and comments on our paper. We have made this correction in the revised version.

*(10). The discussion section summarizes many of the paper's findings and analyses, a more discussion of uncertainties and insights regarding ozone pollution control would be beneficial.*

**Response**: Thank you for your suggestion regarding the discussion section. We have revised the manuscript in the following two ways:

**a. Uncertainty Analysis**: The primary source of uncertainty in the spatiotemporal analysis lies in the long-term ozone estimates. To address this, we conducted rigorous validation of the historical ozone estimates, in contrast to many previous studies that relied solely on sample-based 10-fold CV to represent the accuracy of long-term estimates. Our validation results, particularly the time-aggregated validation outcomes, demonstrate significant improvements in the accuracy of the historical estimates, with an $R^2$ of 0.74 at the monthly scale in Fig. 1. These findings suggest that the spatiotemporal exposure analysis, especially regarding long-term variations, is robust and reliable. This discussion has been incorporated into the final paragraph of Section 4 in the revised version. Below is the revised content from our manuscript.

"The primary source of uncertainty in this study lies in the long-term ozone estimates. Since the NAQMN was not established before 2013, monitoring data from earlier years is unavailable. As a result, we could not directly train the model for that period. Instead, we applied the model developed for post-2014 data to hindcast ozone levels for the earlier unmonitored years. Consequently, the estimated ozone levels for these years may carry a certain degree of uncertainty, which could impact the spatiotemporal analysis. However, we conducted rigorous validation of the hindcast estimates, and the time-aggregated validation results demonstrated significant improvements in the accuracy of the pre-2013 estimates (R2 of 0.74 at the monthly scale in Fig. 1). These findings suggest that the spatiotemporal exposure analysis, particularly regarding long-term variations, is robust and reliable."

**b. Challenges in Controlling Ozone Pollution**: We have included ozone pollution control recommendations based on our spatiotemporal analysis in the middle paragraphs of Section 4 in the revised manuscript. For instance, we observed a notable shift in the peak ozone exposure month from June to May, which is particularly pronounced in the NCP region. We discussed the potential causes of this phenomenon and recommended that policymakers remain vigilant about this shift to adapt mitigation strategies in the third paragraph of

Section 4. Additionally, we have included a paragraph in the fourth paragraph of Section 4 to discuss the implications of our county-level study on policymaking.

---

## Author Comment (AC2)

**Response to Reviewer 2:**

*Comments:*

*The study analyzed the long-term and short-term characteristics of surface ozone spatialtemporal variation and exposure in 2000-2020, which provides important support for environmental management and health research. This study used a machine learning method based on satellite retrieved surface temperature, combined with multi-scale and multi-temporal dimensions analysis, to reveal the spatial and temporal distribution of ozone and its potential health risks. This detailed and comprehensive study have high scientific value, and provide a reference basis for policy making in related fields. However, there are still some aspects that need to be further improved.*

**Response**: Thank you for reviewing our manuscript and providing us constructive suggestions/comments. We have thoroughly considered your suggestions/comments and revised our manuscript accordingly. The item-by-item responses are below.

Major:

*(1). The background of the article is not well researched, and the introduction only briefly mentions some O3 data at high spatial resolution for O3. However, there are already studies (Shang et al., 2024. https://doi.org/10.1021/acs.estlett.4c00106) where hourly O3 data are available, and the advantages of your research over these studies should be fully explained.*

**Response**: Thank you for sharing the publication focusing on ground-level ozone modeling with high temporal resolution. We have reviewed this paper and incorporated a discussion in the second paragraph of Section 1 in the revised manuscript, as shown below. Overall, while previous studies, including the one you referenced, have made significant advancements in improving spatial and/or temporal resolutions, they have yet to achieve high resolution in both dimensions simultaneously, such as daily 1-km estimates—an area where this study offers a distinct advantage.

"… While previous studies have estimated ground-level ozone concentrations across China with improved estimation performance (Ma et al., 2022a; Liu et al., 2020; Wei et al., 2022a; Zhan et al., 2018; Chen et al., 2021; Shang et al., 2024), at least two key limitations persist: (1) despite advancements in resolution, these studies have not achieved high resolution in both spatial and temporal dimensions, such as daily 1-km estimates. …"

*(2). There is too much restatement in the discussion section and the conclusion section, I suggest that the language be refined and that the two sections, or some of them, be considered to be merged together.*

**Response**: Thank you for your suggestion. We have revised the discussion and conclusion sections to reduce redundancy between them. Specifically, in the revised discussion section, the content on NCP from the initial version has been integrated into the third paragraph, while the second paragraph now focuses solely on the national, long-term, three-phase trend in ground-level ozone. In the updated conclusion section, we summarize the key findings of this study in three aspects: (1) improved ozone hindcasting using satellite-derived LST, (2) non-monotonous long-term trends and seasonal shifts, and (3) emerging exposure hotspots.

*(3). Why aerosol optical depth used as an index for O3 retrieval? As far as I know, solar radiation is also an important factor in the photochemical reaction of O3. Will it be improved if it is included as a feature in the model for training?*

**Response**: The reasons for including AOD as a predictor in our ground-level ozone estimation model are outlined below. This information has been incorporated into Section S1.1 of the revised supporting document.

> ➢ Atmospheric chemical and physical perspective: Aerosols influence the scattering and absorption of solar radiation, thereby affecting the efficiency of photochemical reactions (Wang et al., 2019), which play a critical role in the formation and destruction of ozone. Additionally, certain aerosols, such as black carbon, can adsorb volatile organic compounds (VOCs) in the atmosphere (Gao et al., 2018), which are important precursors for ozone formation. By altering VOC concentrations, aerosols can indirectly impact ozone production. AOD, as a primary proxy for aerosol physical properties and concentrations, was therefore incorporated as a relevant predictor in our model.

> ➢ **Statistical perspective**: Previous studies have demonstrated a correlation between aerosol particles and ozone levels (Zhu et al., 2022). Based on this relationship, AOD was included as a predictor in our ozone estimation model. Our variable importance analysis (Fig. S9) ranked AOD 7th among the 15 predictors in the machine-learning model, further highlighting its contribution to ground-level ozone modeling.

Regarding solar radiation, we agree that it is a critical factor in ozone formation, particularly due to the direct influence of ultraviolet light on ozone generation. To account for the effects of solar radiation, we included two relevant variables in our estimation model: sunshine duration (SSD) and surface solar radiation downwards (SSRD). As shown in Fig. S9, SSRD ranked second only to land surface temperature (LST) in terms of variable

importance for predicting long-term MDA8 ozone concentrations using the proposed method.

Minor:

*(1). Figure 3 lacks the label of (b).*

**Response**: We would like to thank the reviewer for pointing out this issue. We have made the necessary modifications to Figure 3 by adding the label (b) as suggested.

*(2). Line 231 has an extra set of parentheses after the NCP.*

**Response**: Thank you for your careful review and comments on our paper. We have noted the writing error in line 231 where "NCP)" should be corrected to "NCP." We have made this correction in the revised version to ensure the accuracy of the paper.